# The Performance Analysis of PSO-ResNet for the Fault Diagnosis of Vibration Signals Based on the Pipeline Robot

**DOI:** 10.3390/s23094289

**Published:** 2023-04-26

**Authors:** Zhaotao Yu, Liang Zhang, Jongwon Kim

**Affiliations:** 1School of Mechanical, Electrical and Information Engineering, Shandong University, Weihai 264200, China; 2WeiHai Research Institute of Industrial Technology of Shandong University, 180 Wenhua Xilu, Shandong University, Weihai 264209, China; 3Department of Electromechanical Convergence Engineering, Korea University of Technology and Education, Cheonan-si 31253, Republic of Korea

**Keywords:** PSO-ResNet, vibration signals, sensors, fault diagnosis, pipeline robot

## Abstract

In the context of pipeline robots, the timely detection of faults is crucial in preventing safety incidents. In order to ensure the reliability and safety of the entire application process, robots’ fault diagnosis techniques play a vital role. However, traditional diagnostic methods for motor drive end-bearing faults in pipeline robots are often ineffective when the operating conditions are variable. An efficient solution for fault diagnosis is the application of deep learning algorithms. This paper proposes a rolling bearing fault diagnosis method (PSO-ResNet) that combines a Particle Swarm Optimization algorithm (PSO) with a residual network. A number of vibration signal sensors are placed at different locations in the pipeline robot to obtain vibration signals from different parts. The input to the PSO-ResNet algorithm is a two-bit image obtained by continuous wavelet transform of the vibration signal. The accuracy of this fault diagnosis method is compared with different types of fault diagnosis algorithms, and the experimental analysis shows that PSO-ResNet has higher accuracy. The algorithm was also deployed on an Nvidia Jetson Nano and a Raspberry Pi 4B. Through comparative experimental analysis, the proposed fault diagnosis algorithm was chosen to be deployed on the Nvidia Jetson Nano and used as the core fault diagnosis control unit of the pipeline robot for practical scenarios. However, the PSO-ResNet model needs further improvement in terms of accuracy, which is the focus of future research work.

## 1. Introduction

Currently, the world is facing pollution problems, and oil pollution is a growing concern. The handling technology for oil pollution is not very mature yet [1]. As the world is moving towards automation, integration, intelligence, and greening, robotic smart devices are being used more and more in various industries. Since the birth of industry, pipelines have been very important tools as they are used in many different industries for various applications such as transporting oil, gas, water, etc. Many defects can occur in pipelines, most of which are caused by aging, corrosion, cracks, blockages, and mechanical disasters that cause harm to human life and the environment. Therefore, the inspection of pipelines is extremely important to improve the reliability and safety of the industry. Currently, one of the most feasible solutions for the inspection of pipelines is the use of pipeline robots [2,3].

Pipeline robots are used for inspection work of large LPG pipelines, aiming to achieve efficient inspection of LPG pipelines and prevent safety accidents [4]. Since pipeline robots are mainly used in unmanned, high humidity, and other extreme and harsh conditions, the safety and reliability of the robots are particularly important. Pipeline robots have more complex structures, more degrees of freedom, and unstable motion accuracy, all of which bring efficiency and safety problems to the application of pipeline robots. Once a failure occurs, it is difficult and costly to repair, and collisions caused by motion problems can easily cause fires, resulting in unnecessary economic losses. The research of fault detection and diagnosis technology has become the focus of developing pipeline robots.

Traditional fault detection and diagnosis methods for pipeline robots include using various physical and chemical principles and means to detect faults directly through various physical and chemical phenomena that accompany faults; using the signs corresponding to faults to diagnose faults, such as using the vibration of fault signals and their spectral characteristics to infer the type of faults [5,6]. Although these methods are fast and very effective, they can only detect some of the faults.

Modern intelligent fault detection and diagnosis methods for pipeline robots are mainly based on the above-mentioned traditional diagnosis methods, and theories and methods of artificial intelligence are used for fault diagnosis to develop intelligent diagnosis methods. Intelligent fault diagnosis has been able to train learning algorithms from a large amount of feature data and diagnostic results. Deep learning algorithms, mainly represented by neural network learning, are now widely used, and neural networks include convolutional neural networks and recurrent neural networks [5,6]. The research of deep learning algorithms has become the main direction of equipment fault diagnosis.

Highly integrated single-board computers can perform tasks such as fault detection and diagnosis independently, and the operation of a pipeline robot requires a processing unit that is responsible for processing digital images and implementing some of the pipeline robot’s functions. Therefore, compact portable single-board computers are essential to support the functionality and performance of pipeline robots. The Raspberry Pi 4B and Nvidia Jetson Nano are two representative portable single-board computer devices, and the Raspberry Pi 4B and Nvidia Jetson Nano have been widely used in a number of real-world scenarios. They are widely used in a number of real-world scenarios.

The main research of this paper is to improve the existing fault diagnosis model as a way to improve the accuracy and generalization performance of the model. In addition, a suitable fault detection and diagnosis algorithm is deployed to test the performance of the algorithm in different single-board computers. Through comparative experimental analysis, the best-performing single-board computer is embedded into the pipeline robot as the control unit for pipeline robot fault detection and diagnosis. The motivation and solution of the research in this paper can be summarized as follows.

(1) As to ensure the pipeline robot works properly in the pipeline, the pipeline robot should have the function of automatic fault detection and diagnosis. The traditional fault detection and diagnosis algorithm cannot be intelligent; through researching various studies, the fault detection algorithm PSO-ResNet is proposed. This is a deep learning fault detection and diagnosis algorithm that combines particle swarm optimization algorithm and ResNet-50.

(2) In the practical application of pipeline robots, the time and accuracy of fault detection and diagnosis and the influence of the temperature of the single-board computer on the operation of the algorithm are important indicators to ensure the safe work of pipeline robots in pipelines. In this paper, the PSO-ResNet algorithm is deployed on a single-board computer for comparison experiments. The time and accuracy of algorithm execution on different single-board computers and the temperature of the single-board computers are compared. The metrics are evaluated together to determine the Nvidia Jetson Nano as a fault detection and diagnosis unit for pipeline robots.

## 2. Related Works

Along with the continuous development of deep learning-based machine fault detection and diagnosis algorithms, more and more researchers are investigating the effective use of deep learning techniques on embedded platforms. In the study, while Raspberry Pi can provide lower power consumption and energy-efficient performance, it can also be concluded that Nvidia Jetson platforms such as Jetson Nano, Jetson TX1, and Jetson TX2 have higher performance because they have higher-speed GPUs [7]. The Nvidia Jetson is considered the most widely used gas pedal in the extraction phase of machine learning. In addition, the Nvidia Jetson Nano is energy efficient and portable, making it easy to use and has high performance in low-power situations [8].

The high-performance CPU and CUDA-compatible GPU make the Nvidia Jetson Nano development board perform well in benchmark tests. From the low-power perspective, the Nvidia Jetson Nano has higher performance, and the Raspberry Pi seems less suitable for high-performance deep-learning algorithm research [9].

Edgar Ruiz compared the performance metrics such as algorithm execution time, algorithm accuracy, CPU performance, and temperature of Raspberry PiCM4 and Nvidia Jetson Nano by using the SVM algorithm. In that study, only machine learning algorithms were compared. The complexity of the algorithms is relatively low and is not studied for more advanced and complex deep learning algorithms [10]. Gunawan Dewantoro used image processing and computer vision techniques to allow the patrol robot to identify objects and control robot motion. The performance of the patrol robot using different CPU controllers (Raspberry Pi and Jetson Nano) was compared. It was concluded that the GPU embedded in the Jetson Nano significantly affected the performance of processing video and graphics, so the robot with the Jetson Nano could perform faster image processing. However, only the accuracy of the algorithms was compared in this study, and the impact of factors such as single-board computer temperature and CPU occupancy on the performance of the single-board computer was not considered [11].

Xiaocui Hong compares the performance of single-board computers Nvidia Jetson Nano, Nvidia Jetson TX2, and Raspberry Pi 4 by using deep learning CNN algorithms. The parameters for performance analysis were defined as consumption (GPU, CPU, RAM, Power), algorithm accuracy, and single-board computer cost, and the data sets for training and testing of the models were classified as 5K, 10K, 20K, 30K, and 45K. Such an operation extends the variability of single-board computer performance [12]. There was no experimental study of algorithms for machine fault detection and diagnosis in this study, and the accuracy of the two-dimensional DNN algorithm for fault detection and diagnosis in pipeline robots was not high [13].

Traditional machine learning fault detection and diagnosis algorithms, such as k-NN-based, plain Bayesian-based, SVM-based, and ANN-based, are relatively low in complexity and easy to implement [5]. However, these traditional machine learning fault diagnosis methods have low recognition rates, poor robustness, and rely on manual feature extraction [14]. Deep learning provides an effective method to automatically learn features at multiple levels of abstraction so that complex input-to-output functions can be learned directly from data without relying on feature extractors, which can be of great benefit for fault diagnosis in pipeline robots [5].

Fault detection and diagnosis algorithms represented by deep learning such as BP neural networks, DNN, CNN, residual networks, and attention mechanism residual networks [13,14,15]. Deep learning networks do not require any feature extraction techniques and have high accuracy and robustness in noisy environments [16,17].

Kai Zhang improves the residual network. A hybrid attention improved residual network (HA-ResNet) based method is proposed to diagnose the fault of wind turbines gearbox by highlighting the essential frequency bands of wavelet coefficients and the fault features of convolution channels [12]. Xiaocui Hong proposed a new method for fusion diagnosis based on a multimodal residual network (M-ResNet) and discriminant correlation analysis (DCA). The M-ResNet consisting of one-dimensional ResNet (1D-ResNet) and two-dimensional ResNet (2D-ResNet) is constructed to extract richer and more comprehensive fault features [13]. Long Wen studied a migrating convolutional neural network for fault diagnosis based on ResNet-50. ResNet-50 trained in ImageNet is used as a feature extractor for fault diagnosis by using a migration learning approach [18]. Kai Zhang proposed a novel method based on an adaptive loss-weighted meta-residual network (ALWM-ResNet). It is proposed to address fault diagnosis with noisy labels using a weighted network and a meta-network cloned from the original ResNet to establish a weighted function mapping to adaptively learn weights from data with clean labels [19]. Shaofeng Fu proposed a SE-ResNet PV array fault diagnosis algorithm based on a residual network (ResNet) and squeezed excitation network (SENet) with Bayesian optimization (BO) for parameters [20]. Xinjie Peng proposed a fault diagnosis method based on a combination of PCA and ResNet. The original data are denoised, and then the data are diagnosed and classified using a residual network. The practicality and robustness of the method were experimentally demonstrated [21]. As the amount of data on the Internet grows, deep learning models take advantage of big data processing. Deep residual networks with higher feature extraction and generalization capabilities are widely used in the field of machine fault detection and diagnosis.

Based on the above research, we propose the PSO-ResNet algorithm model. This model can actively find the optimal hyperparameters in network training compared to other fault diagnosis models. It provides researchers with a more scientific approach to training the model. We also deployed PSO-ResNet to Nvidia Jetson Nanoh and Raspberry Pi 4B. We tested the performance of embedded devices through comparative experiments. The search for more price-performance devices is important for the development of pipeline robots.

## 3. Description of the Proposed Method

The overall process of this study is divided into three parts: signal acquisition and input, single-board computer fault diagnosis, and fault result output. The input signals used in the pipeline robot fault detection and diagnosis study are vibration signals, and the single-board computer is mainly Nvidia Jetson Nano and Raspberry Pi 4B. The fault diagnosis results are the types of faults occurring in the pipeline robot. The fault detection and diagnosis algorithm use a residual network combining continuous wavelet transform, particle swarm optimization algorithm, and ResNet-50. The main block diagram of the proposed method is shown in Figure 1.

### 3.1. Pipeline Robot

The main work of pipeline robots is to inspect oil pipelines, obtain abnormal locations, and prevent LPG pipeline safety accidents. The pipeline robot can enter the pipeline in a relatively narrow space and replace the workers to complete the inspection and maintenance of the pipeline, which can greatly improve the efficiency of pipeline maintenance and reduce maintenance costs. The pipeline robot has a reasonable ontology structure and control method to ensure that it does not slip, overturn, or jam inside the pipeline; the pipeline robot can accurately identify the early and minor faults that appear in itself to avoid the minor faults from evolving into serious faults that cause the robot to lose control and fail to exit the pipeline. Figure 2 shows the working situation of the pipeline robot.

### 3.2. Nvidia Jetson Nano

Nvidia Jetson Nano is a small, powerful computer for embedded applications and AI IoT that delivers the power of modern AI. It has the comprehensive JetPack SDK with accelerated libraries for deep learning, computer vision, graphics, multimedia, and more. Meanwhile, the Nvidia Jetson Nano can run multiple neural networks in parallel for applications such as image classification, object detection, segmentation, and speech processing. All on an easy-to-use platform that runs in as little as 5 watts. Figure 3 shows the Nvidia Jetson Nano with 2 GB RAM. The official Nvidia website shows a comparison between the performance of this single-board computer and the Raspberry Pi 3, Intel Neural Compute Stick 2, and Google Edge TPU Coral development motherboard, where the Jetson Nano outperforms the aforementioned platforms [10].

### 3.3. Raspberry Pi 4B

The Raspberry Pi 4B is a microcomputer the size of a credit card. It has great speed and performance improvement over the earlier models. It has many features, such as editing documents, opening a bunch of tabs to browse the web, working with spreadsheets, and editing presentations. The biggest advantage of the Raspberry Pi 4B is that it is smaller, more energy efficient, and more economical. The overall architecture is shown in Figure 4.

### 3.4. PSO-ResNet Model

The continuous wavelet transform is developed on the basis of the Fourier transform [22]. The continuous wavelet transform can solve the problem that the Fourier transform exists where the time domain and frequency domain information of the signal cannot be localized at the same time. Real-world data or signals often exhibit a tendency to change slowly or oscillate due to transients [23]. On the other hand, images have smooth regions that are interrupted by edges or sudden changes in contrast, and the Fourier transform cannot effectively represent sudden changes because the Fourier transform represents data as a sum of sinusoids that are not localized in time or space, and these sinusoids oscillate forever [24,25]. In order to analyze abruptly changing signals and images well and accurately; the wavelet transform is used that is well localized in both time and frequency. Wavelet transforms are rapidly decaying waves and wavelets exist for a finite duration [26].

Particle Swarm Optimization (PSO) is a global optimization algorithm based on population intelligence. In the PSO algorithm, each solution is considered as a position of a particle, while each particle represents a candidate solution, and the optimal solution is searched by continuously updating the velocity and position of the particles. Specifically, each particle maintains two vectors: a position vector and a velocity vector [27]. During the search process, the position and velocity of each particle are continuously adjusted until the optimal solution is found or a preset stopping condition is reached. Figure 5 illustrates the overall flow of the particle swarm optimization algorithm. Particle swarm optimization algorithms are simple to implement and do not require many parameters to be adjusted. In general, it can be applied to a variety of complex problems to find globally optimal solutions with fast convergence and high efficiency. Currently, particle swarm optimization algorithms have been widely used in function optimization, neural network training, fuzzy system control, and other applications of genetic algorithms [28].

Optimization of residual network models using particle swarm optimization algorithms. The main use of the particle swarm optimization algorithm is to optimize the initialization parameters of the training model [27]. The optimization parameters are set for the particle swarm optimization algorithm, which includes: the number of particles, the maximum and minimum values of the parameters being optimized, the number of iterations of the optimization algorithm, and the maximum velocity of each particle per movement [29]. The particle swarm optimization algorithm can search for the optimal solution and find the best parameters for the network training in order to be used in the training of the residual network model. In addition, the particle swarm optimization algorithm is applied to optimize the weights and biases of each residual block in ResNet-50 in order to improve the performance of the network [30]. In each iteration, the PSO algorithm can be used to update the position and velocity of each particle in order to find the best solution. By continuously iterating, the optimal weights and deviations can be found, thus optimizing the performance of the residual network. The particle swarm optimization algorithm is combined with the residual network, which not only effectively improves the network model generalization ability and accuracy but also increases the speed at which the network model learns the differences between different faults and effectively distinguishes between impassable fault types [28].

As a modified convolutional network, the residual network is based on a series of convolutional and pooling layers [16]. Convolutional layers achieve sparse interaction and parameter sharing during data transfer, reducing the number of model parameters [19]. ResNet implements a layer backoff mechanism by adding a Skip Connection between the input and output of the convolutional layers. Input x is passed through two convolutional layers to obtain the feature-transformed output F(x), which is summed with the corresponding elements of input x to obtain the final output H(x):(1)H(x)=x+F(x),
H(x) is called the residual module [31]. The residual structure is shown in Figure 6.

In the study of this paper, we used the ResNet structure shown in Figure 7, which was adopted for two main reasons: (1) the backward propagation basically conforms to the assumption that the information transfer is unhindered; (2) the BN layer acts as a pre-activation and plays a role of regularization. This has a great effect on the network training to improve the accuracy of the model.

As shown in Figure 8, PSO-ResNet mainly consists of the signal processing part and the ResNet-50 network learning part. ResNet-50 is mainly divided into 5 stages, in which the structure of Stage 0 is relatively simple and can be regarded as the pre-processing of the input, and the last 4 stages are composed of Bottleneck, which have a similar structure. Stage 1 contains 3 Bottlenecks, and the remaining 3 stages include 4, 6, and 3 Bottlenecks, respectively.

The optimization algorithm for residual neural networks determines the optimal value of the parameters by minimizing the error in the cost function, which in turn improves the performance of the network. The choice of learning rate and parameter optimization algorithms for residual neural networks will affect the performance of the network to some extent for different problems. In the training of residual networks, the optimization of hyperparameters plays a very important role in the speed of network training as well as the robustness and extensiveness of the model. The learning rate is a hyperparameter that controls the extent to which the residual network adjusts the network weights in terms of loss gradient; momentum is a common acceleration technique used in gradient descent methods. A suitable momentum can achieve an accelerated convergence process and facilitate our training [28].

During the training of the PSO-ResNet, if the learning rate is set too large, the network will not converge and will hover around the optimum, thus neglecting to find the optimum; if the learning rate is set too small, the network will converge very slowly and will increase the time to find the optimum. Although it is possible to achieve better training results by setting a very small learning rate, this is likely to converge into the local extrema, with no real optimal solution found. In this paper, the initial value of the learning rate when first training the PSO-ResNet model is 0.00001. In general, the learning rate ranges from 0.00001 to 0.1. As the number of iterations increases, the learning rate decreases to accelerate the convergence of the loss function. Throughout the training period, the learning rate from the beginning to the end of training is reduced by a factor of one hundred of the initial value.

According to the optimization idea of the particle swarm algorithm, the optimal hyperparameters are used as the search target of the particle swarm algorithm. The loss value is the fitness function value, and the parameters corresponding to the best fitness value are recorded. In each iteration, the particles are continuously optimized in the direction of these two parameters, and the resulting new parameters are sequentially fed into the network and iteratively updated until the end of the iteration.

In this paper, training experiments of the PSO-ResNet network are conducted on a homemade dataset, and the values of the loss function of the network under the same number of iterations are used as the individual fitness scores. After multiple iterations, the local optimum and the global optimum are obtained, and their corresponding hyperparameters are saved.

The biggest advantage of PSO-ResNet is that it not only reduces the cost for researchers to tune hyperparameters and accelerates the training of ultra-deep neural networks but also significantly improves the accuracy of deep networks. In addition, PSO-ResNet largely avoids the problem of gradient disappearance or gradient explosion as the number of layers of the network increases, which makes it possible to train extremely deep networks. 

## 4. Experimental Study and Results

Jetson Nano and Raspberry Pi as the core control unit for pipeline robot fault diagnosis. Due to the complex architecture of the PSO-ResNet network, training the model with two single-board computers are almost impossible. The environment for model training is a 64-bit Windows 11 operating system, the CPU is 13th Gen Intel(R) Core(TM) i7-13700F 2.10 GHz, the graphics card is NVIDIA Ge Force RTX 3060, and the memory is 16.00 GB. The entire algorithm is written in Python, and the model framework is built by PyTorch.

The motor bearing faults are set as follows: damage of 0.1778 mm diameter, damage of 0.3556 mm diameter, and damage of 0.5334 mm diameter. The types of faults are shown in Table 1. In order to have more data to participate in model training, based on the Case Western Reserve University (CWR) bearing data, we placed some vibration signal sensors above the bearing seat at the motor drive end of the pipeline robot to collect the vibration signal of the faulty bearing. In addition, Jetson Nano and Raspberry Pi were embedded in the pipeline robot to receive the collected vibration signals, and the structure of the pipeline robot is shown in Figure 9.

The trained algorithm models are deployed on two single-board computers. In addition, the performance metrics considered for the single-board computer benchmark are algorithm accuracy (%), runtime (ms), CPU usage (%), and temperature (°C).

### 4.1. Handling of Fault Data

Before the model training, the acquired fault data are first processed. The acquired signals are converted into 2D images by continuous wavelet transform, and these 2D images are used as inputs for the deep learning algorithm to train the model. There are 1000 2D images for each class of faults, and the proportions of each class of faults as a training set, test set, and validation set are 70%, 20%, and 10%, respectively, during the training.

### 4.2. Model Deployment

For the Nvidia Jetson Nano, to accelerate inference, the network is deployed using the TensorRT framework, a neural network inference framework that is optimized to speed up inference and reduce the model size, and deploy neural networks on edge computing devices for significant computational efficiency [32]. Converting network models created by Jetson Nano or PC using Pytorch framework to Open Neural Network Exchange (ONNX) format, then converting to TensorRT format using ONNX parser, and deploying the converted TensorRT format to Jetson Nano for inference is an efficient and popular approach.

For the deployment of the model on Raspberry Pi 4B, since Raspberry Pi does not contain the TensorRT inference framework, the network model created using the Pytorch framework is directly converted to Open Neural Network Exchange (ONNX) format, converted to IR intermediate model using openvino, downloaded openvino on Raspberry Pi, and used the IR model to perform inference.

### 4.3. Analysis of Experimental Results

After the pipeline robot detects a fault, the single-board computer acts as the core control unit for fault detection and diagnosis of the fault signal output. The two single-board computers display the type of fault through a connected monitor, and the main results and categories of the output are shown in Figure 10.

#### 4.3.1. The Performance Comparison of Different Fault Diagnosis Models

First, the experiments were conducted to compare the fault diagnosis accuracy for different fault diagnosis models separately. The fault diagnosis models CNN, VGG16, and ResNet-50 are selected for the accuracy comparison analysis with the fault diagnosis model PSO-ResNet proposed in this paper, and the comparison results are shown in Table 2. The fault diagnosis model proposed in this paper has a large improvement in accuracy. Compared with ResNet-50 without the particle swarm optimization algorithm, PSO-ResNet improves accuracy by about 2%. 

For residual neural networks, the setting of hyperparameters is one of the most important factors in determining the performance of the model. Hyperparameters such as momentum and learning strategy will affect the accuracy of algorithm detection. The improved fault kind detection algorithm PSO-ResNet has a lower training error rate as well as a generalization error rate than the original algorithm, which reflects the algorithm’s strong ability to optimize the neural network connection weights and to seek a better learning rate. Moreover, in terms of training time, the training time of PSO-ResNet with the addition of particle swarm optimization algorithm does not lag much behind the training time of ResNet-50, and the training speed is also relatively fast. Obviously, PSO-ResNet, with higher accuracy and faster training speed, is more suitable as a fault detection algorithm for the fault diagnosis system of the pipeline robot.

#### 4.3.2. The Parameters Comparison of Jetson Nano and Raspberry Pi

The Raspberry Pi 4B has a quad-core Cortex-A72 (ARM v8) 64-bit CPU, while the Nvidia Jetson Nano has a quad-core ARM Cortex-A57 MPCore CPU, which gives the Raspberry Pi 4B faster processing speed in terms of CPU performance. However, the Raspberry Pi 4B has a Broadcom VideoCore VI GPU (32-bit), while the Nvidia Jetson Nano has an NVIDIA Maxwell GPU w/128 CUDA cores. In terms of GPU performance, the Nvidia Jetson Nano has a faster image processing speed. A comparison of the specific parameters is shown in Table 3. 

#### 4.3.3. The Performance Comparison of Jetson Nano and Raspberry Pi

Before testing the fault diagnosis algorithm, some performance attributes inherent to the two single-board computers were tested. Table 4 shows the performance of the two single-board computers in terms of data transfer, and Table 5 shows the temperature and CPU usage of the two single-board computers in standby mode. Since the pipeline robot stores the fault data in the storage unit of the single board computer first when it receives it, it is necessary to test the transfer rate.

To obtain the performance characteristics of CPU and temperature, SystemMonitor of each single board computer was used to obtain these two types of data. Regarding accuracy and execution time data, these metrics were obtained by executing a fault diagnosis algorithm. The test of fault types was divided into 1 fault signal at a time input, 10 fault signals, and 100 fault signals. Testing the depth algorithm model based on PSO-ResNet, the program was run 100 times, and the results of the runs were averaged on the Nvidia Jetson Nano and Raspberry Pi 4B. Table 6 shows the data obtained from testing the PSO-ResNet model.

In this context, as seen in the results shown in Table 5, both single-board computers provide similar algorithmic accuracy, indicating that this parameter depends more on the algorithm than on the performance of the single-board computer.

As far as the execution time of the algorithm is concerned, it can be observed that the values of the Raspberry Pi 4B and the Nvidia Jetson Nano are about the same when one sample signal is input. The main reason for this phenomenon is that the number of input samples is small, and the single-board computer uses the CPU more for processing, while the CPU of the Raspberry Pi is a bit higher in performance than the Jetson Nano. When inputting 10 sample signals, the average execution time of Nvidia Jetson Nano is 152.33 ms faster than Raspberry Pi 4B, while when processing 100 input sample signals, the average execution time of Nvidia Jetson Nano is 15,154.4 ms faster than Raspberry Pi 4B, Jetson Nano executes about four times faster than the Raspberry Pi 4B. This difference is very surprising but reasonable because the Nvidia Jetson Nano has a better-performing GPU, which is helpful for more computationally intensive signal processing. In addition, it is very important for the safe and reliable operation of the pipeline robot to be able to know in time if a failure occurs.

Regarding the percentage of CPU used, similar to the algorithm execution time, the number of incoming fault signals gradually increased at any time, and the CPU occupancy of the Raspberry Pi became higher and higher, even exceeding 60%. Although the Nvidia Jetson Nano has an elevated CPU occupancy, the CPU idle state is still sufficient, which ensures a smooth-running state of the single-board computer. In a similar situation, the Nvidia Jetson Nano requires fewer CPU resources to perform the same task.

## 5. Conclusions

In this paper, a new fault diagnosis model PSO-ResNet is investigated for pipeline robot fault diagnosis, and the performance of different control units in the pipeline robot fault diagnosis module is analyzed. PSO-ResNet is a deep evolutionary model for finding the best combination of hyperparameters in residual network training. The excellent performance of the particle swarm algorithm is used to find the most suitable hyperparameters to indirectly improve the performance of the network model. This method not only reduces the labor cost for researchers to manually adjust hyperparameters but also makes it more scientific to adjust hyperparameters in network training. The experimental results show that the PSO-ResNet fault diagnosis algorithm model has a large improvement in accuracy, and the model has a better generalization ability and can find the best weights and deviations to optimize the performance of the residual network by selecting better parameters for the training of the model. When tested on a single-board computer, although the PSO-ResNet fault diagnosis algorithm model has the same accuracy on different single-board computers, the Nvidia Jetson Nano has better performance in terms of execution time, temperature, and CPU usage. In addition, the deployment of the fault diagnosis algorithm to the embedded GPU system Nvidia Jetson Nano using the TensorRT framework significantly reduces the inference time. In other ways, the smaller CPU footprint and better performing GPU means that the single-board computer can also do more work, which is helpful in studying pipeline robots to implement model updates for newly generated faults and expand more useful features of pipeline robots. Finally, while the Nvidia Jetson Nano is a bit more expensive when using deep learning algorithms for complex network models to accomplish specific tasks, the Nvidia Jetson Nano’s superior performance makes it the preferred choice for low-cost embedded systems for pipeline robots.

Although the PSO-ResNet model has achieved good results in experiments, there is still room for improvement in terms of accuracy. In order to improve the model’s accuracy, future research can focus on improving the model’s architecture and fine-tuning the pre-trained network model for optimization. Additionally, the PSO-ResNet model occupies a large amount of memory, which limits its application in resource-limited environments such as mobile devices and embedded systems. Therefore, lightweighting the PSO-ResNet model is also one of the future research priorities.

## Figures and Tables

**Figure 1 sensors-23-04289-f001:**
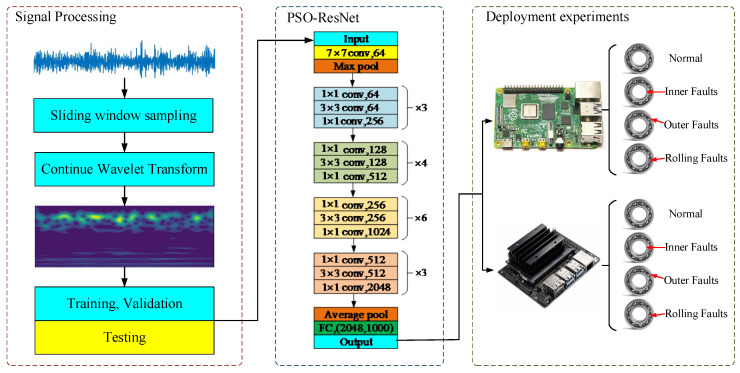
The overall framework proposed in this study.

**Figure 2 sensors-23-04289-f002:**
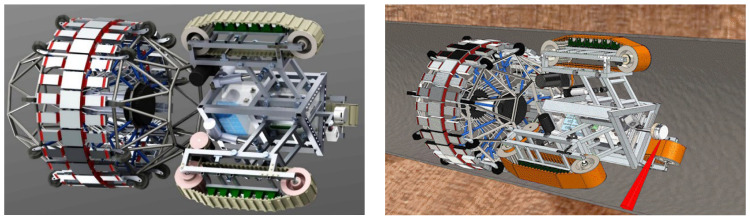
Pipeline robot structure and operation in the pipeline (See Appendix A for dynamic version).

**Figure 3 sensors-23-04289-f003:**
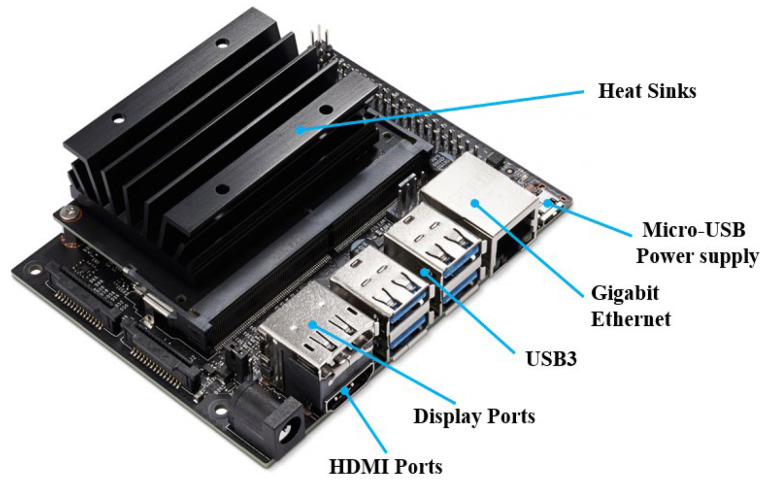
The structure of the Nvidia Jetson Nano.

**Figure 4 sensors-23-04289-f004:**
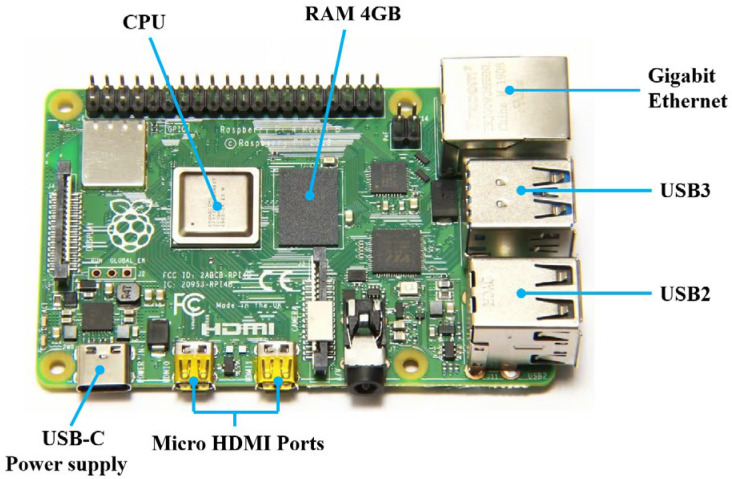
The structure of the Raspberry Pi 4B.

**Figure 5 sensors-23-04289-f005:**
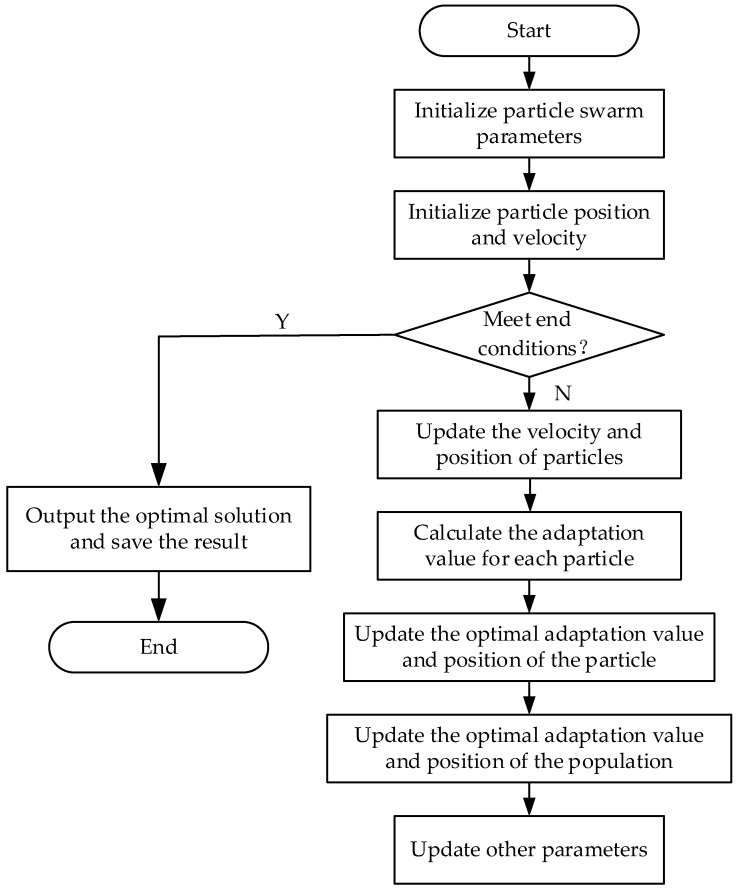
The overall flow of the particle swarm optimization algorithm.

**Figure 6 sensors-23-04289-f006:**
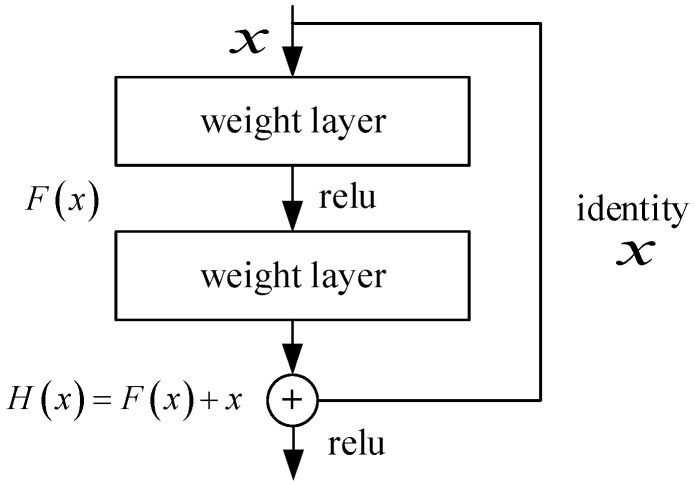
Schematic diagram of the structure of the residual module.

**Figure 7 sensors-23-04289-f007:**
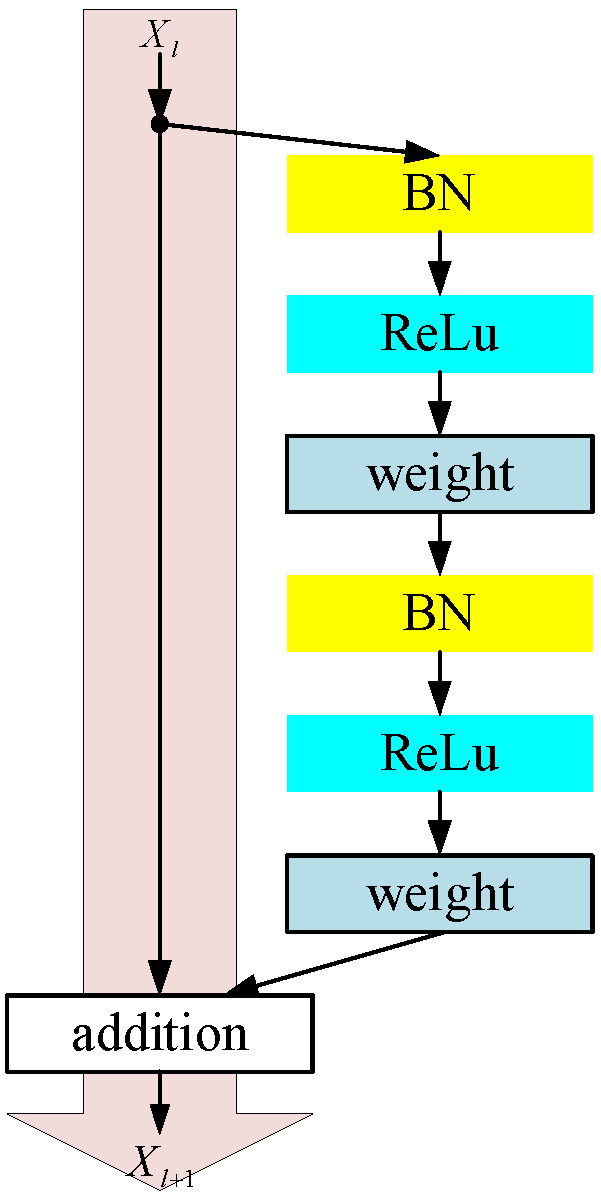
The diagram of the ResNet structure used in this study.

**Figure 8 sensors-23-04289-f008:**
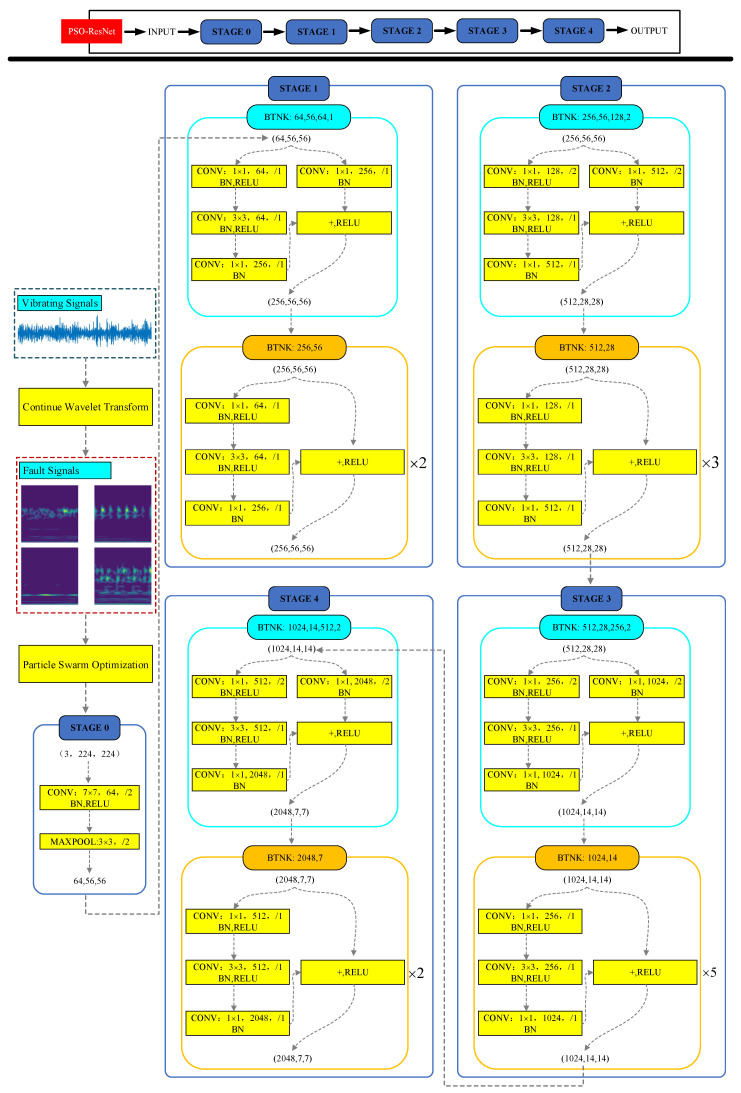
The Schematic diagram of the overall structure of PSO-ResNet.

**Figure 9 sensors-23-04289-f009:**
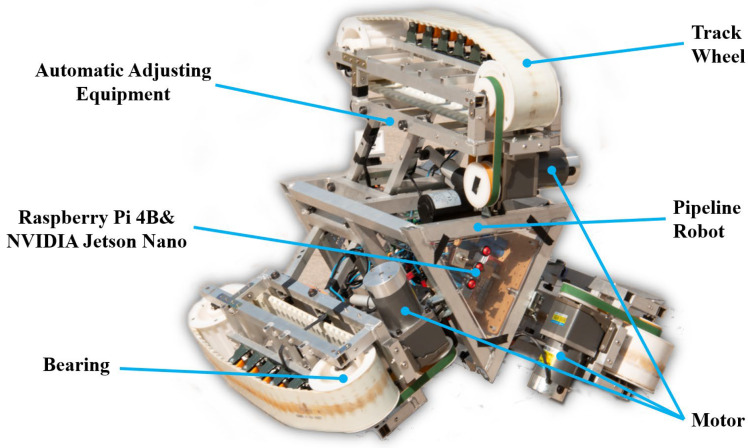
The schematic diagram of the structure of the pipeline robot.

**Figure 10 sensors-23-04289-f010:**
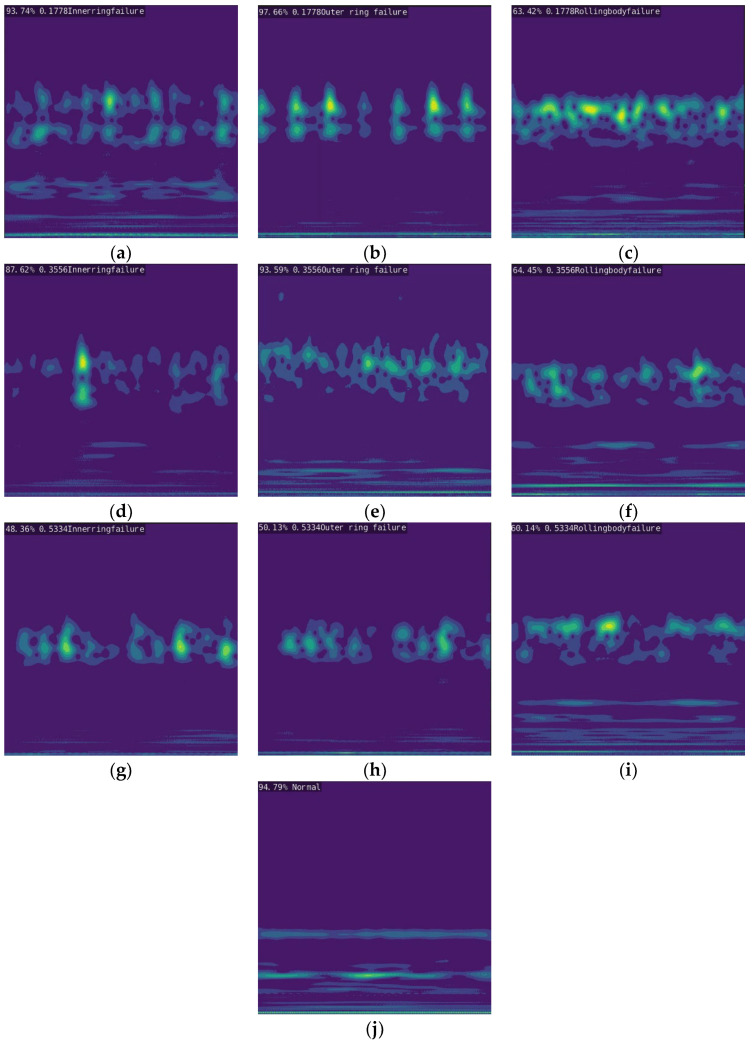
The result of the fault diagnosis output performed by the pipeline robot. (**a**) 0.1778 Inner Ring Failure; (**b**) 0.1778 Outer Ring Failure; (**c**) 0.1778 Rolling Body Failure; (**d**) 0.3556 Inner Ring Failure; (**e**) 0.3556 Outer Ring Failure; (**f**) 0.3556 Rolling Body Failure; (**g**) 0.5334 Inner Ring Failure; (**h**) 0.5334 Outer Ring Failure; (**i**) 0.5334 Rolling Body Failure; (**j**) Normal.

**Table 1 sensors-23-04289-t001:** The types of pipeline robot bearing failures.

Diameter of Bearing Damage (mm)	Damaged Location	Label
0.1778	Inner ring failure	0
	Outer ring failure	1
	Rolling body failure	2
0.3556	Inner ring failure	3
	Outer ring failure	4
	Rolling body failure	5
0.5334	Inner ring failure	6
	Outer ring failure	7
	Rolling body failure	8
/	Normal	9

**Table 2 sensors-23-04289-t002:** Comparative results.

Fault Diagnosis Model	Accuracy (%)
CNN	90.24
VGG16	94.85
ResNet-50	94.94
PSO-ResNet	96.85

**Table 3 sensors-23-04289-t003:** The parameters comparison of Jetson Nano and Raspberry Pi.

	Raspberry Pi 4B	NVIDIA Jetson Nano
CPU	Quad-core ARM Cortex-A72 64-bit	Quad-Core ARM Cortex-A57 64-bit
GPU	Broadcom VideoCore VI (32-bit)	NVIDIA Maxwell w/128 CUDA cores
Memory	4 GB LPDDR4	4 GB LPDDR4
Networking	Gigabit Ethernet/Wifi 802.11ac	Gigabit Ethernet/M.2 Key E
Display	2 × micro-HDMI	HDMI 2.0 and eDP 1.4
USB	2 × USB 3.0, 2 × USB 2.0	4 × USB 3.0, USB2.0 Micro-B
Other	40-pin GPIO	40-pin GPIO
Video Encode	H264 (1080p30)	H.264/H.265 (4Kp30)
Video Decode	H.265 (4Kp60), H.264 (1080p60)	H.264/H.265 (4Kp60, 2 × 4Kp30)
Camera	MIPI CSI port	MIPI CSI port
Storage	Micro-SD	Micro-SD
Price	$105 USD	$149 USD

**Table 4 sensors-23-04289-t004:** The transfer rate comparison of Jetson Nano and Raspberry Pi.

	USB 3.0 SSD (MB/s)	Micro SD (MB/s)
Jetson Nano	288.4	62.5
Raspberry Pi	265.2	40.8

**Table 5 sensors-23-04289-t005:** The temperature and CPU usage in standby mode of Jetson Nano and Raspberry Pi.

	CPU (%)	CPU Temperatures (°C)	GPU (%)	CPU Temperatures (°C)
Jetson Nano	9.24	29.25	0	29
Raspberry Pi	3	47.225	0	48.50

**Table 6 sensors-23-04289-t006:** The performance of running PSO-ResNet model of Jetson Nano and Raspberry Pi.

		Jetson Nano	Raspberry Pi
Algorithm Precision (%)		96.85	96.85
Algorithm Execution Time (ms)	1 Sample	276.37	250.49
	10 Samples	397.45	549.78
	100 Samples	4772.40	19,926.80
CPU (%)	1 Sample	28. 45	26.90
	10 Samples	37.95	47.74
	100 Samples	42.28	65.45
Temperature (°C)	1 Sample	30.75	49.50
	10 Samples	38.50	62.50
	100 Samples	43.50	65.50

## Data Availability

The data used in the experiments can be found at https://engineering.case.edu/bearingdatacenter/download-data-file.

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
