# Peer review of "The Performance Analysis of PSO-ResNet for the Fault Diagnosis of Vibration Signals Based on the Pipeline Robot"

_sensors, 2023, doi:10.3390/s23094289_

Round 1
Reviewer 1 Report (Previous Reviewer 1)
Can be accepted.
Reviewer 2 Report (Previous Reviewer 2)
No more comments are required.
Reviewer 3 Report (Previous Reviewer 3)
The authors improved the paper as per my comments. So, I recommended
for publication.
This manuscript is a resubmission of an earlier submission. The following is a list of the peer review reports and author responses from that submission.
Round 1
Reviewer 1 Report
In this paper, wavelet transform and CWT-ResNet are used to diagnose the bearing fault of pipeline robot, and the application performance of Jetson nano and raspberry pie is compared. The paper has a certain reference value for engineering applications, but the paper lacks substantive innovation in theory, and the hardware used is not very meaningful. More high-performance hardware can be used. The paper also has obvious deficiencies in the number of fault data samples and experiments.
Reviewer 2 Report
1- L12 – L17 in the abstract needs to be revised in technical writing. Poor and long sentences are noticed.
2- Main methods and findings should be concisely mentioned in the abstract. Authors should consider the following;
a) Add some results regarding the optimum conditions and highlight the novel findings of the study.
b) Highlight the study limitations and future recommendations.
3- It appears that this work is part of a research dissertation. The authors have to rewrite the work objectives and motivations in a continuous flow of phrases.
4- L88 – L93 in the introduction part needs to be revised. No need to write section descriptions.
5- The authors can follow the following reference to generally highlight the global contamination problems in the beginning of the introduction part; (https://doi.org/10.3390/ma15134547). Then, highlight the need to consider the offshore oil fields problem.
6- Section 2 “Related Works” could be merged in the introduction part, then indicate a clear gap in knowledge which this study seeks to bridge, and potentially contribute to knowledge. After that highlight the research objectives.
7- Cost evaluation is a major concern of this study.
8- Does corrosion affect the efficiency/durability of the pipeline robot?
9- The authors have to consider the environmental concerns of the electronic device used in the study.
10- Highlight the limitations, recommendations and future prospects of the study in the conclusion part.
Reviewer 3 Report
, In this paper, the authors have propose a rolling bearing fault diagnosis method based on the combination of Continuous Wavelet Transform and Residual Network (CWT ResNet), which is deployed on Nvidia Jetson Nano and Raspberry Pi 4B. The two embedded hard ware are embedded in the pipeline robot for algorithm testing. The paper is interesting and innovate enough for publication to this journal but some changes are required: My suggestions are as follows:
1) I suggest the authors to improve the introduction section. Authors should better highlight the objective of their work and to what extent it contributes to close a gap in the existing literature and/or practice. What is the innovative value of the contribution proposed by the authors?
2) In introduction section authors should provide more information about existing models used in the field and their benefits/weaknesses. This should be discussed. The authors need to discuss their contributions compared to those methods in related papers.
3) The authors must clearly discuss the significance of the research problem in the first section.
4) You should provide more recent references published in last two-three years. Remove references published before 2017. Also, remove lumped references, all references should be discussed in the manuscript. You must cites suggested papers in algorithmics approach in decision science areas: Some Dombi aggregation of Q‐rung orthopair fuzzy numbers in multiple‐attribute decision making, A robust single-valued neutrosophic soft aggregation operators in multi-criteria decision making; Assessment of enterprise performance based on picture fuzzy Hamacher aggregation operators; A dynamical hybrid method to design decision making process based on GRA approach for multiple attributes problem.